# Comparative Genomics and Evolutionary Analysis of RNA-Binding Proteins of *Burkholderia cenocepacia* J2315 and Other Members of the *B. cepacia* Complex

**DOI:** 10.3390/genes11020231

**Published:** 2020-02-21

**Authors:** Joana R. Feliciano, António M. M. Seixas, Tiago Pita, Jorge H. Leitão

**Affiliations:** 1iBB—Institute for Bioengineering and Biosciences, Instituto Superior Técnico, Universidade de Lisboa, 1049-001 Lisbon, Portugal; antonio.seixas@tecnico.ulisboa.pt (A.M.M.S.); tiagopita@tecnico.ulisboa.pt (T.P.); 2Departamento de Bioengenharia, Instituto Superior Técnico, Universidade de Lisboa, 1049-001 Lisbon, Portugal

**Keywords:** *Burkholderia cepacia* complex, RNA-binding proteins, comparative genomics, Hfq, cold shock proteins, RhlE helicase

## Abstract

RNA-binding proteins (RBPs) are important regulators of cellular functions, playing critical roles on the survival of bacteria and in the case of pathogens, on their interaction with the host. RBPs are involved in transcriptional, post-transcriptional, and translational processes. However, except for model organisms like *Escherichia coli*, there is little information about the identification or characterization of RBPs in other bacteria, namely in members of the *Burkholderia cepacia* complex (Bcc). Bcc is a group of bacterial species associated with a poor clinical prognosis in cystic fibrosis patients. These species have some of the largest bacterial genomes, and except for the presence of two-distinct Hfq-like proteins, their RBP repertoire has not been analyzed so far. Using in silico approaches, we identified 186 conventional putative RBPs in *Burkholderia cenocepacia* J2315, an epidemic and multidrug resistant pathogen of cystic fibrosis patients. Here we describe the comparative genomics and phylogenetic analysis of RBPs present in multiple copies and predicted to play a role in transcription, protein synthesis, and RNA decay in Bcc bacteria. In addition to the two different Hfq chaperones, five cold shock proteins phylogenetically close to *E. coli* CspD protein and three distinct RhlE-like helicases could be found in the *B. cenocepacia* J2315 genome. No RhlB, SrmB, or DeaD helicases could be found in the genomes of these bacteria. These results, together with the multiple copies of other proteins generally involved in RNA degradation, suggest the existence, in *B. cenocepacia* and in other Bcc bacteria, of some extra and unexplored functions for the mentioned RBPs, as well as of alternative mechanisms involved in RNA regulation and metabolism in these bacteria.

## 1. Introduction

RNA-binding proteins (RBPs) are found in all domains of life, playing a critical role in the stabilization, protection, processing, and transport of RNA, as well as in the posttranscriptional control of gene expression [1,2]. RBPs are commonly classified based on their specific RNA binding domains (RBDs), i.e., structural protein domains that directly bind to specific RNA sequences and/or structured domains in RNA [3]. The classical bacterial RBDs include the S1 domain, the cold-shock domain, the Sm and Sm-like domains, the double-stranded RNA binding domain, the K-homology domain, the DEAD motif, and the ANTAR domain. These domains are widely distributed and/or conserved among different bacterial species (previously reviewed [4,5]). However, proteins that do not harbor any conventional direct RNA-binding site [6], but are able to interact with RNA or proteins in a non-classic way (“unconventional” RBPs) have also been described [7].

Ribosomal proteins (r-proteins) are the most abundant and best characterized RBPs that have been identified and annotated in bacterial genomes [5,8]. These proteins, together with other RBP major classes such as tRNA synthetases, RNA helicases, and ribonucleases, are critical for many cellular processes. In addition to their involvement in processes associated with RNA metabolism and protein synthesis, the importance of bacterial RBPs in the extensive control of gene expression at the posttranscriptional level has been highlighted over the past two decades. While some RBPs can regulate transcription termination via attenuation (e.g., Rho, NusA, and the *Bacillus subtilis* TRAP and PyrR proteins) or anti-termination mechanisms (e.g., cold shock proteins, HutP, Bgl/Sac), others can repress or activate translation initiation by affecting ribosome biding or by changing RNA stability [9]. The regulation mediated by RBPs is mainly due to their interaction with small non-coding RNAs (sRNAs) [4]. sRNAs are short, non-coding RNA molecules that can act as global regulators of gene expression in prokaryotes [10,11,12,13]. In order to perform their regulatory activity, sRNAs often require the aid of global RBPs like RNA chaperones, that facilitate their interaction with cognate mRNA targets, affecting numerous physiological processes [14,15]. Our knowledge of global RBPs remains limited to the Hfq chaperone, the translational repressor CsrA, and to the more recently characterized “osmoregulatory” protein ProQ [5,16,17]. This limitation is partially due to experimental difficulties of identifying bacterial RBPs, the advances in understanding these proteins being virtually confined to bioinformatics tools to faithfully predict RNA binders in bacteria [18,19]. Moreover, the current knowledge regarding the number, functions, and mechanisms of the bacterial RBPs remains also scant for non-model microorganisms, as is the case of bacteria of the *Burkholderia cepacia* complex (Bcc). Bcc is a group of at least 24 closely related bacterial species that attracted the attention of various research groups worldwide due to their ability to cause problematic, difficult-to-eradicate, and often fatal infections among cystic fibrosis patients [20,21,22,23]. In addition, recent reports also mention an increasing number of infections caused by these bacteria in non-cystic fibrosis patients, including hospitalized patients suffering from other malignancies such as cancer, hemodialysis, and others [24,25,26,27]. These bacteria possess large genomes arranged in multiple replicons with high plasticity and complex regulatory mechanisms of gene expression [28]. Our research group has previously reported that two distinct Hfq-like RNA chaperones are encoded in the genomes of Bcc bacteria, the 79 amino acid residue Hfq, and the 189 amino acid residue Hfq2 [29]. 

Besides the Hfq-like proteins, scarce studies are available on RBPs in Bcc bacteria. Therefore, in the present work we report a bioinformatics survey and comparative genomics analyses to identify “conventional” RBPs within the genomes of Bcc. The genome of the highly epidemic strain *B. cenocepacia* J2315 was used as reference since it is one of the best studied Bcc strains [23]. The sequences of the identified RBPs, especially those that differed in number of copies per genome, were retrieved from available genome sequences, and their phylogenetic and evolutionary relationships within the *Burkholderia* genus were analyzed. The predicted functions of these proteins were compared with the *Escherichia coli* homologs, revealing that in addition to two distinct Hfq-like proteins, five cold shock-like CspD proteins, and three distinct RhlE-like helicases are encoded in *B. cenocepacia* and in other Bcc genomes. This study contributes to unveil putative protein partners of posttranscriptional regulation in pathogens of the Bcc, suggesting that undisclosed mechanisms should be used by these bacteria to regulate their gene expression.

## 2. Materials and Methods 

### 2.1. Database Searches for Putative RBPs in B. cenocepacia J2315 Genome

Putative homologs of RBPs were retrieved from public domain databases using a multipronged search approach. First, the canonic RBDs found widespread among bacterial RBPs were used as keywords in Pfam [30], InterPro [31], and Uniprot databases to search for putative RBPs in *B. cenocepacia* J2315 genome. The canonical RBDs used were the S1 domain, the cold-shock domain, the Sm and Sm-like domains, the double-stranded RNA binding domain, the K-homology domain, the DEAD motif, the ANTAR domain, the zinc-finger like domain, the PIWI domain, and the PAZ domain.

As a second strategy, proteins from the best known model organism *E. coli* and the opportunistic pathogen *Pseudomonas aeruginosa*, classified as “RNA binding” according to Gene Ontology, were used to search for homologs against the proteome of *B. cenocepacia* J2315. *E. coli* and *P. aeruginosa* were used as reference since both species present the best annotated prokaryotic genomes. Briefly, proteins from the taxons *E. coli* strain K-12 (83333) and *P. aeruginosa* PAO1 (208964) annotated with a RNA binding related GO term were retrieved from the fast web-based browser Quick GO (Gene Ontology Annotation database) [32]. The proteins retrieved were annotated with the GO terms GO:0003723 (RNA binding), GO:0019843 (rRNA binding), GO:0000049 (tRNA binding), GO:1903231 (mRNA binding involved in post-transcriptional gene silencing), GO:0034336 (misfolded RNA binding), GO:0003727 (single-stranded RNA binding), GO:0003729 (mRNA binding), GO:0003725 (double-stranded RNA binding). After obtaining the ortholog genes between *E. coli* (strain K-12) MG1655 and *B. cenocepacia* J2315 genomes, or between *P. aeruginosa* PAO1 and *B. cenocepacia* J2315 genomes, using the OrtholugeDB predictions (Ortholog Database version 2.1) [33], the protein-coding genes previously selected based on GO terms search were selected and the respective orthologs in *B. cenocepacia* J2315 genome were retrieved. These data were further confirmed searching for orthologs of each initially selected protein in EggNOG [34] and KEGG [35] databases. The genes sequences retrieved from the above analyses were combined and parsed to remove duplicated genes, to obtain the final list of putative RBPs encoded within the *B. cenocepacia* J2315 genome.

### 2.2. Putative RBPs in Bcc and Non-Bcc Burkhoderia Genomes

Search of orthologs for each selected RBP, whether or not encoded on *B. cenocepacia* J2315, was performed using KEGG and *Burkholderia* genomes databases, and the genomes of Bcc and non-Bcc species listed in Appendix A [36]. The number of homolog RBPs found among the genomes analyzed, as well as the percentage of *Burkholderia* species that contain them are shown in Appendix A. The phyletic profile of each putative RBP-coding gene listed in Appendix A was obtained based on data provided by OrthoDB [37], the hierarchical catalog of orthologs. Using the RBP names as keywords, the gene orthologs “at Bacterial level” were selected and the evolutionary descriptions were accessed.

### 2.3. Multiple Sequence Alignments and Phylogenetic Analyses

The nucleotide and amino acid sequences of the RBPs analyzed in the present study and listed in Appendix A were retrieved from the GenBank database at the National Center for Biotechnology Information (NCBI) [38], and also from the Burkholderia Database [36]. To find homologs of the identified RBPs, non-redundant database searches were performed by both standard protein-protein BLAST (BLASTp) and Position-Specific Iteractive BLAST (PSI-BLAST) against the Swiss-Prot [39] and eggNOG 4.5 [34] databases with a cut-off *E*-value ≤0.0001. These results were further compared with the information available on KO (KEGG Orthology) database [40].

Except for phylogenetic trees of Figures 3, 4 and 6, all the sequence alignments were performed using the MUSCLE (Multiple Sequence Comparison by Log-Expectation) [41] web server with default settings, for both the nucleotide and amino acid sequences, and edited with Jalview [42], to extract the common core of conserved residues. The set of sequences used in each alignment are indicated on figure legends. Protein secondary structure predictions were performed using the Protein Structure Prediction Server (PSIPRED) [43]. Maximum-likelihood phylogenetic trees were generated by MEGA X [44] using the best predicted substitution model for each group of aligned sequences, and 150 bootstrap replications. The aligned sequences and the phylogenetic trees of the functional categories COG1923 (Conserved Protein Domain family Hfq), COG1278 (Cold Shock Proteins), and COG0513 (Superfamily II DNA and RNA Helicases), containing, respectively, 798, 4636 and 9254 amino acid sequences from organisms from different phyla, were downloaded from EggNOG database [34]. In addition to generate the ortholog groups, EggNOG uses a combination of tools and workflows implemented in the Python ETE tree building tool for the reconstruction of phylogenetic trees. The resulting trees in Newick format were drawn and annotated in the Interactive Tree Of Life web server (iTOL) [45].

### 2.4. Evolutionary Analysis 

The Ka/Ks ratio, where Ka is the number of non-synonymous substitutions per non-synonymous site and Ks is the number of synonymous substitutions per synonymous site, was used as an indicator of selective pressure acting on the protein-coding gene. The Ka/Ks values were calculated using the KaKs_Calculator 1.2 [46], employing the Nei–Gojobori method and computing pairwise distances through the MEGA X software. Analyses were performed among orthologs of *hfq*-like, cold shock-like, and *rhlE*-like genes using only the Bcc sequences or all the sequences of the 24 *Burkholderia* and 2 *Paraburkholderia* species that are listed, respectively, in Appendix A. The Ka/Ks values were also calculated among the *hfq* (*hfq* and *hfq2*) and *rhlE* (*rhlE1*, *rhlE2* and *rhlE3*) paralogs. 

### 2.5. Statistical Analysis

Statistical analysis was performed using Prism 6 (GraphPad) software (San Diego, CA, USA). Two-tailed Mann–Whitney test (non-parametric) was used. Data obtained were represented as mean ± standard deviation (S.D.). Results with a *p*-value < 0.05 were considered significant. 

## 3. Results and Discussion

### 3.1. Putative “Conventional” RBPs Identified in the B. cenocepacia J2315 Genome

Using a combination of strategies involving comparative genomics and search for classical RBDs, 186 putative “conventional” RBPs were identified in the *B. cenocepacia* J2315 genome, as listed in Appendix A. According to their functional category, the 186 proteins were distributed as follows: 45 putative ribosomal proteins (large and small subunits); 25 rRNA modification factors; 24 tRNA modification factors; 23 aminoacyl-tRNA synthetases; 13 ribosome associated proteins; six termination/antitermination factors; 11 translation factors; 14 ribonucleases and accessory factors; six helicases; 14 RNA chaperones/regulators; and five other proteins, two of which involved in RNA degradation (Appendix A). Since we are particularly interested in RBPs involved in posttranscriptional processes, the proteins that modulate the termination phase of transcription (for both induction and prevention of termination) were also included in this analysis. However, both RNA polymerases and transcription factors, whose role in RNA-binding is well documented, were excluded.

Except for an RNase, the SpoU protein, and proteins containing the ANTAR domain, homologs genes of putative RBPs identified in *B. cenocepacia* J2315 are also encoded in the *E. coli* genome. Nonetheless, using the approach described, no sequence based, or functional homologs could be found in *B. cenocepacia* J2315 genome for 41 RBPs encoded by the *E. coli* and/or *P. aeruginosa* genomes. More than half of these missing proteins were functionally categorized as rRNA or tRNA modification factors that are not widely distributed by bacterial species (less than 30%) or have multiple copies in the *E. coli* genome (Appendix A). It is important to mention that several modification factors, corresponding to enzymes of the same type and annotated with the same OrthoDB reference, are differently classified regarding the COG and/or functional category, hindering the analyses carried out (Appendix A). Most of the other *E. coli* RBPs without orthologs in *B. cenocepacia* J2315 genome can only be found in a limited percentage of bacterial species, and in some cases, they seem to be order or family specific: the asparaginyl-tRNA synthetase (AsnS) is an aminoacyl-tRNA synthetase present in ≈50% of bacteria; only 17.11% of bacterial species encode the BglG anti-terminator; 7.34% the ribosome modulation factor Rmf; 35.45% the ribosome associated inhibitor A (RaiA); 6.67% the alternative ribosome rescue factor (ArfA); 13.08% the regulator of ribonuclease activity B (RraB); 18.65% the StpA chaperone; and 34.69% the carbon storage regulator (CsrA). CsrA is a global regulator of post-transcriptional gene expression that has been widely studied in both *E. coli* and *P. aeruginosa*. Except for a *B. pseudomallei* strain, no CsrA orthologs were found in Bcc genomes or in any other strain of the *Burkholderia* genus, suggesting that alternative global regulators should be encoded in the genomes of these bacteria.

From the nine *E. coli* toxins identified as RNA-binding proteins, only orthologs of the endoribonuclease toxin MazF and the toxic protein SymE were found in the genome of *B. cenocepacia* J2315. Despite the relatively low distribution of the identified *E. coli* toxins among the bacterial genomes (0.45–40%), the toxin-antitoxin (TA) systems are generally widespread as a consequence of their capacity to be transferred horizontally [47,48]. In addition, 16 pairs of genes were previously identified in *B. cenocepacia* J2315 based on their apparent similarity to TA modules, although it is unknown if any RNA-binding protein is present in these systems [49]. Based on this, broader research was carried out, covering the specific motifs present in each toxin. Except for the mRNA interferase toxin RelE that seems to share a motif with BCAL0070, no match was found.

The accurate assessment of the presence of highly conserved and multi-copy proteins’ orthologs across species, such as the helicase class or the cold shock family, could be challenging. Therefore, together with the identified ribonucleases, both helicases and cold shock proteins were subjected to a more detailed phylogenetic analysis (see sections below).

Contrasting with the absence of some previously identified *E. coli* RBPs in the available *Burkholderia* genomes, about 20 putative RBPs were identified in multiple copies in *B. cenocepacia* J2315. Gene gain and gene duplication events have been recognized as a prominent factor in the evolution of the large multireplicon genomes from bacteria of the *Burkholderia* genus [50]. This characteristic is even more peculiar on chromosome 1 of *B. cenocepacia* J2315, since it contains a perfectly duplicated region of ≈57 kb, including genes BCAL0969 to BCAL1026 and BCAL2901 to BCAL2846 [51]. Several proteins that participate in a diversity of functions were reported to be encoded in these duplicated region [52], including proteins that display RNA-related functions. For instance, ribosomal proteins (RpmF), rRNA modification factors (RluC and RsmI), ribonucleases (RNase E and III), and Era GTPases are proteins that interact directly with RNA and are encoded by duplicated genes, while GroEL, TufA, elongation factors, and the GTP protein LepA are proteins involved in RNA related processes, such as the RNA degradation process or the protein synthesis. The increased gene dosage hypothesis seems to underlie this perfect duplication mechanism. However, the identified homologs genes are not always the result of a perfect duplication event as mentioned above. In addition to RNA helicases and cold shock proteins, the small subunit ribosomal protein S21 (RpsU), the 16S rRNA modification factor RsmI, the methionyl-tRNA synthetase MetG, and the RNA chaperone Hfq are encoded by distinct homolog genes. In order to assess the conservation and evolution of these homolog genes among bacteria of *Burkholderia* genus, especially those of Bcc, *in silico* predictions and phylogenetic analysis were performed.

### 3.2. R-proteins and Other RBPs Involved in Protein Synthesis in B. cenocepacia J2315 and in Other Bacteria of the Burkholderia Genus

RBPs involved in protein synthesis, such as r-proteins, ribosome-associated proteins, tRNA synthetases, and enzymes that modify tRNA and/or rRNA, are widely distributed among bacteria and are the largest functional class of bacterial RBPs. The bacterial 70S ribosome results from the assembly of the 50S large subunit, containing 23S rRNA, 5S rRNA, and about 33 r-proteins (named L1 to L36), and the 30S small subunit, that is composed of 16S rRNA and about 22 proteins (S1 to S22) [53]. Although duplication of bacterial r-protein-encoding genes is a rare event, r-proteins L9, L20, L27, L35, S6, S20, and S22 seem to never duplicate in bacterial genomes, while the number and the identity of paralogs of other r-proteins are variable among different groups of bacteria [54]. Regarding the disposition of r-proteins in the 30S ribosome subunit, in *E. coli* a set of proteins bind primarily and independently to the 16S RNA, being followed by the assembly of S5, S9, S12, S13, S16, S18, and S19 proteins that, finally, seem to potentiate the addition of S2, S3, S10, S11, S14, and S21.

The *B. cenocepacia* J2315 genome harbors 23 small subunit and 34 large subunit r-protein encoding genes without obvious paralogs, except for genes encoding the large subunit protein L32 (RpmF), and the small subunit protein S21 (RpsU) (Figure 1A,B; Appendix A). The RpmF paralogs BCAL0990 and BCAL2880 are identical (ID = 100%), resulting from their location on the perfectly duplicated ≈57 kb region mentioned above. Distinctively from the duplicated RpmF paralogs, the RpsU paralogs are encoded as three distinct copies by genes *BCAL0115*, *BCAM0915*, and *BCAS024* located, respectively in chromosomes 1 and 2, and in the megaplasmid pC3. The protein sequence of the RpsU paralogs encoded in chromosomes 1 and 2 are 76% identical and share an identity percentage (ID) of 60% or 57%, respectively, with the paralog encoded on megaplasmid pC3 (Figure 1C). The relative rates of synonymous and nonsynonymous substitutions were estimated among the sequences of the RpsU paralog genes, and the calculated values were below 1 (Ka/Ks < 0.26), suggesting a selective pressure to conserve these protein sequences. Results from a previous study have shown that under reduced oxygen concentration these three homolog genes are differentially regulated in *B. cenocepacia* J2315. While BCAL0115 and BCAS0245 are both downregulated, BCAM0915 is slightly upregulated [55]. In addition, the expression of BCAL0115 seems also to be dependent of the growth phase, BCAM0915 responds to low pH; and BCAS0245 probably is involved in maintaining the translation at lower temperature [55]. In *E. coli*, S21 was identified as the smallest and most basic protein of the 30S ribosomal subunit, and it seems also to be one of the few proteins that exchanges *in vivo* between ribosomes [56]. Comparing with other r-proteins, S21 can only be found in a small fraction of bacteria [54] (Appendix A). However, when present in a genome, several paralogs can be found, distributed across different partitions of the bacterial genome. For instance, a strain of *Rhizobium leguminosarum* harbors four S21 paralog copies, two encoded on the major chromosome, and the other two on different plasmids [54]. Despite the assumptions performed, the biological relevance of these S21 multiple copies for *Burkholderia* species needs to be further investigated.

The ribosomal small subunit RNA methyltransferase I (RsmI) protein is a rRNA modification factor that catalyzes the 16S rRNA 2′O-methylation in *E. coli*. The protein is active *in vitro* on the assembled 30S subunit, but not on naked 16S rRNA or on the 70S ribosome [57]. A single copy of this gene is present in most bacterial genomes. However, *B. cenocepacia* J2315 harbors three genes encoding RsmI-like proteins, *BCAL0160*, *BCAL0987*, and *BCAL2883*. Genes *BCAL0987* and *BCAL2883* result from the already mentioned perfectly duplicated region present in chromosome 1, sharing 100% identity. Remarkably, these proteins share only 27.35% identity with BCAL0160, which is evolutionarily closer to the *E. coli* RsmI protein (ID = 48.9%). Common domains can be found in the three putative RsmI-like proteins of *B. cenocepacia* J2315, such as the tetrapyrrole methylase superfamily domain, but the SAM-dependent methyltransferase RsmI conserved site is only present in BCAL0160. These results suggest that BCAL0987 and BCAL2883 are not RsmI-like proteins, and their putative methyltransferase activity most probably results from their ability to accept methyl groups from a methyl donor alternative to SAM.

As described for RsmI, most bacterial genomes harbor a single copy of the MetG methionyl-tRNA synthetase, and only 8% of the bacterial genomes harbor multiple copies of this enzyme. The enzyme recognizes an initiator tRNA of protein synthesis, as well as the tRNA delivering methionine for elongation of protein chain. Even among strains of the *Burkholderia* genus, a second copy of this gene seems to be encoded only in ≈2% of the sequenced genomes. The BCAL2646 and BCAS0045 gene products from *B. cenocepacia* J2315 were identified as MetG-like proteins, containing a methionyl/leucyl-tRNA synthetase domain, and a Rossmann-like alpha/beta/alpha sandwich fold. These two proteins only share 27.72% identity in their first 200 N-terminus amino acid residues. In addition, the *B. cenocepacia* J2315 BCAL2646 is the ortholog of the *E. coli* MetG protein, sharing an identity of 52.7%. It has been described that contrasting with most of other aminoacyl-tRNA synthetases, MetG shows structural diversity among species, most probably related with its ability to interact with other proteins to form functional complexes [58]. In the non-Bcc species *Burkholderia thailandensis* encoding a single copy of this synthetase, a reduced growth rate, and an increased tolerance to meropenem and other antibiotics with different modes of action was observed for a *metG* mutant [59]. The importance of MetG in the resistance to antimicrobial compounds is a possible driving force to maintain two distinct copies of genes encoding MetG-like proteins in the *B. cenocepacia* J2315 genome.

### 3.3. Contrasting With Hfq2, Hfq is Highly Conserved Among Bcc and is an Ortholog of Hfq Proteins From Other β- and γ-Proteobacteria

In a previous work we have described that contrasting with the majority of Gram-negative bacteria, members of *Burkholderia* genus (more recently split in two genera, the genus *Burkholderia*, englobing mainly animal and plant pathogens, and the genus *Paraburkholderia,* including the non-pathogenic and so-called environmental species [60]) encode in their genomes two distinct Hfq-like proteins, the 79–80 amino acid residues Hfq, and the 175–216 amino acid residues Hfq2 [29]. However, it is still unclear why two different Hfq-like proteins are encoded and maintained in the genomes of Bcc bacteria. Here, the conservation and evolution of the Hfq and Hfq2 homologs from 24 *Burkholderia* and 2 *Paraburkholderia* species were analyzed. Using publicly available genome sequences, 26 *hfq* and 26 *hfq2* nucleotide sequences were aligned using the bioinformatics tool MUSCLE. The inferred phylogenetic tree revealed that the *hfq* and the *hfq2* sequences are clearly separated in two well-defined clades (Figure 2A). In addition to the obvious 327 nucleotides length difference at their 3’-terminus for *B. cenocepacia* J2315, the nucleotide composition in the 5′-terminal region is also distinguishable among the *hfq* and *hfq2* sequences, which share about 58% sequence identity. While the Hfq encoding gene is highly conserved among Bcc and other *Burkholderia* species (nucleotide sequences ID > 92%, protein sequences ID > 96%), a lower identity is shared by Hfq2 sequences of Bcc bacteria (ID > 83%), and also of other species of the *Burkholderia* genus (ID > 55%) (Figure 2A1, Figure 2).

The evolution of both *hfq* genes was also evaluated based on nonsynonymous (Ka) and synonymous (Ks) substitutions rates, which were calculated among orthologs from Bcc, or including orthologs from other *Burkholderia* species. The average Ka/Ks values for the *Burkholderia hfq* genes were very low (<0.01), suggesting that the gene is evolutionarily highly conserved (Figure 2B). The Ka/Ks values for *hfq2* were significantly higher than for *hfq* (*p* < 0.0001), but still close to 0.1 for Bcc *hfq2* orthologs, and approximately 0.30 for orthologs from non-Bcc species of *Burkholderia* and *Parburkholderia* genera. Despite being a conserved gene, especially among Bcc bacteria, results suggest that *hfq2* is suffering a weak selective pressure (Figure 2B). However, the selection acting on *hfq2* is not constant over the full-length coding sequence. While the structured N-terminus region of Hfq2 is well conserved (ID_nucleotide_ > 84%), showing very low Ka/Ks values (<0.05), the unstructured C-terminus shows a greater variability (ID_nucleotide_ > 55%) and higher Ka/Ks values (≈0.3) (Figure 2C). A mucin-like domain is present in this C-terminus extension, which is composed of a glycine-rich region containing several repetitive motifs. The lack of tertiary structure in this region could confer a great flexibility to the Hfq2 protein, allowing the establishment of multiple interactions with distinct ligands such as RNA, DNA, other proteins, or lipids, as has been described for other intrinsically disordered RBPs [61]. To further explore the evolutionary relationships of both *hfq* and *hfq2* genes, a phylogenetic tree was reconstructed using the amino acid sequences of 798 Hfq proteins from 765 species, and a strategy similar to the one described by Huerta-Cepas and co-authors [62] (Figure 3). While the Hfq evolution seems to reflect the organismal evolution, being clustered together with Hfq proteins of other Beta (β) and Gamma (γ)-proteobacteria, the Hfq2 homologs were clustered in a distinct group but relatively close to Hfq proteins found in the γ-proteobacteria members *Methylobacter tundripaludum*, *Methylomonas methanica*, *Methylococcus capsulatus*, and some *Legionella* species. This result suggests the likelihood of paralog relationships between the *hfq* and *hfq2* genes, since the duplicated gene grouped in a clade relatively close of the original copy. Similarly, two distinct Hfq-like proteins, possibly sharing paralog relationships, were also found to be encoded in the genomes of four species from the Deferribacteraceae family, whose Hfq amino acid sequences were used to infer the phylogenetic tree represented on Figure 3. This was also previously reported by Sun and co-authors, whose phylogenetic analysis produced no evidence for lateral transfer of Hfq proteins [63].

Although the notorious divergence among the *Burkholderia hfq* and *hfq2* sequences, Ka/Ks analysis were performed among the two genes of the 26 *Burkholderia* and *Paraburkholderia* species selected and mentioned before. The calculated values were higher than 1, indicating that *hfq* and *hfq2* genes are subject to positive selection. 

### 3.4. Cold Shock Proteins Encoded in Bcc Genomes 

The family of cold shock proteins (Csps) is a well-defined set of small and conserved RNA chaperone proteins that commonly play a role in low temperature adaptation. Although the phylogeny of the Csps has been recently deeply analyzed and reclassified in Bacteria, so far most of these proteins have been named following the convention used for *E. coli*, which harbors nine cold shock proteins, named CspA to CspI [64]. In *E. coli*, CspE and CspC are constitutively expressed at physiological temperatures, while CspD is induced under nutrient stress, and CspA, CspB, CspG, and CspI are highly induced after cold shock [65]. In *B. cenocepacia* J2315, the typical cold shock domain, which adopts a five-stranded antiparallel beta-barrel structure, was found in five distinct proteins, three of them encoded on chromosome 1 (BCAL0368, BCAL2732, and BCAL3006), and the other two on chromosome 2 (BCAM1619 and BCAM1810). These proteins are highly conserved, especially on regions containing the RNA-binding motifs, sharing an identity percentage of at least 73%. In order to gain further insights into the evolution of these proteins and try to find out homolog genes in other bacteria, 4636 sequences of cold shock proteins from the three domains of life (annotated with the COG1278 reference) were aligned and used to reconstruct the phylogenetic tree presented on Figure 4. Interestingly, the *B. cenocepacia* J2315 Csps were clustered together with the cold shock proteins of the other members of the *Burkholderia* genus, and with Csps of members of the β-proteobacteria class, suggesting paralog relationships among the genes that encode these proteins (Figure 4, orange cluster). This well-defined cluster is relatively close of a set of cold shock proteins that includes the *P. aeruginosa* PA2622 and the *E. coli* CspD proteins. The proteins grouped in these two clusters were included in the Clade III of the recently revised classification of the bacterial Csps in five clades and 12 subclades [64]. Contrasting with the other defined clades, where *csp* genes in the same bacterial class were clustered together, the *cpsD* genes of γ-proteobacteria in Clade III clustered together with the Csp genes of the β-proteobacteria, indicating a common evolution.

In addition to inhibit DNA replication, the *E. coli* CspD is induced when cells enter into the stationary-growth phase or when exposed to nutrient starvation [66,67]. The protein is involved in biofilm development and in persister cell formation, and its overproduction seems to be toxic for bacterial cells [68,69]. Since the Csps encoded by *B. cenocepacia* J2315 exhibit a sequence ID ranging from 55% to 65% with the *E. coli* CpsD (Figure 5B), it is expected that these proteins have some common functions. However, the presence of five distinct cold shock-like proteins sharing precisely the same function is unlikely, even for bacteria with a characteristic genetic redundancy like *Burkholderia* spp. This hypothesis is supported by the expression levels of these five genes that were previously monitored in *B. cenocepacia* J2315 under different growing conditions [55]. Similar to CspD, BCAL3006 is induced in minimal medium when cells enter into the stationary-growth phase. In rich medium, BCAL2732 is differentially expressed in stationary phase, but instead of being induced, is downregulated [55]. Although the expression of cold shock-like proteins was expected to be activated at low temperature, comparing with growth at 37 °C only BCAL0368 and BCAM1619 transcripts are induced at 20 °C. On the other hand, BCAL3006 was found to be more expressed at 37 °C and after a short-term exposure of the cells to 42.5 °C [55]. In addition, BCAL3006 is downregulated under oxidative stress conditions, while the other *B. cenocepacia* J2315 Csps are downregulated under a low concentration of oxygen.

To gain a more comprehensive view on the conservation and phylogeny of the cold shock-like proteins encoded by members of the *Burkholderia* genus, 135 nucleotide sequences from 24 *Burkholderia* (18 belonging to Bcc) and two *Paraburkholderia* species were selected and analyzed in more detail. Four to six cold shock-like proteins are encoded in the genomes of Bcc bacteria, while three to eight are encoded in genomes of other species of the *Burkholderia* genus, or in *Paraburkholderia*. The phylogenetic tree inferred from the multiple sequence alignment of the selected nucleotide sequences revealed a clear separation of the *Burkholderia* cold shock-like proteins in five phylogenetic groups, corresponding to the five Csp members encoded in the majority of the *Burkholderia* species (the names of the five *B. cenocepacia* J2315 *csp* genes were used to name each cluster; Figure 5A).

The average of Ka/Ks values were estimated for BCAL0368, BCAL2732, BCAL3006, BCAM1619, or BCAM1810 orthologs. The orthologs grouped on BCAL2732 and BCAL3006 clusters are more conserved and phylogenetically more closely related (Figure 5A,C). This is especially interesting considering that both genes showed a growth-phase dependent expression. The Ka/Ks values were significantly higher for the BCAM1619 orthologs (Figure 5C). Remarkably, paralog genes of strains that encode more than five Csps in their genomes also clustered with BCAM1619. Considering the selective pressure to conserve the sequences of these cold shock-like proteins, and despite their similarity to the *E. coli* CspD, it is expected that some specific but yet unknown functions are being performed by each *B. cenocepacia* Csp. This hypothesis is also supported by the distinct expression patterns of these genes [55].

### 3.5. DEAD-Box RNA Helicases and Proteins Involved in RNA Degradation 

RNA helicases of the DEAD-box family are a group of highly conserved enzymes with orthologs in many bacteria. These proteins participate in a variety of processes involving RNA, including RNA decay. Typically, these proteins contain nine conserved motifs involved in the ATPase and helicase activities, as well as in the regulation of these activities, and some of them have also been reported to participate in RNA binding [70]. *E. coli* contains five extensively studied DEAD-box proteins (DEAD, DbpA, RhlB, RhlE, and SrmB), whose main functions are summarized in Table 1. 

Five putative DEAD-box proteins (BCAL2117, BCAM0173, BCAL0933, BCAM2814, and BCAL2412) are encoded in the *B. cenocepacia* J2315 genome. In contrast with *E. coli*, three of these proteins are distinct homologs of the RhlE protein, and no *E. coli* RhlB or SrmB homolog was found in Bcc genomes (Appendix A). To confirm these data and to better understand the phylogeny of the DEAD-box helicases in *Burkholderia* species, a phylogenetic tree based on 9254 sequences of proteins classified as having a putative purine NTP-dependent helicase activity (COG0513 annotation) from 1815 species of all domains of life was reconstructed as shown in Figure 6. *B. cenocepacia* J2315 helicases BCAL0933, BCAL2412, and BCAM2814 from *B. cenocepacia* J2315 split into three different clusters of the clade containing the *E. coli* RhlE protein (b0797) and the *P. aeruginosa* PA0428 and PA3950 helicases. BCAL0933 protein is evolutionarily closer related to the *E. coli* RhlE (ID = 66.9%) and *P. aeruginosa* PA0428 (ID = 62.63%) proteins. On the other hand, BCAM2412, which is encoded on chromosome 2 and present in a lower percentage of *Burkholderia* strains (≈44%, contrasting with the wide distribution of the other two *rhlE* homologs genes), shares an identity of 70.3% with the *P. aerugionosa* PA3950 protein. BCAM2412 is also the *B. cenocepacia* J2315 *rhlE* homolog that shares a lower identity with the *E. coli* RhlE protein (ID = 55.4%) (Figure 6). The presence of several additional DEAD-box proteins clustering in groups closely related to *E. coli* RhlE was previously reported for a few bacterial species, like those of the *Shewanella* genus belonging to γ-proteobacteria [84,85]. However, the reason why multiple copies are present in the genome is still unknown, since the inactivation of the *rhlE* gene by itself does not seem to influence growth or ribosome biogenesis. A previous study reported that in a *csdA* mutant (*deaD* gene), the RhlE protein was identified as a multicopy suppressor of the cold-sensitive phenotype [81,86]. This is quite interesting considering that no DeaD homolog is present in the *B. cenocepacia* J2315 genome.

To analyze the evolution of these three RhlE-like proteins identified in *B. cenocepacia* J2315, 24 orthologs of each encoding gene were retrieved from the genome of other species of the *Burkholderia* genus. The inferred phylogenetic tree and the calculated Ka/Ks values suggest that *BCAM2814* (*rhlE3*) gene orthologs are significantly more conserved (lower Ka/Ks ratio) than the *BCAL2412* (*rhlE2*) or the *BCAL0933* (*rhlE1*) gene orthologs (Figure 7A,C). While the sequences of RhlE3 encoding genes are highly conserved in the analyzed species of the *Burkholderia* genus, the identity shared by the ortholog sequences of RhlE1 and RhlE2 is higher among Bcc species and decreases among other members of the *Burkholderia* genus (Figure 7A). These results suggest that the RhlE1, as well as the RhlE2 orthologs, are diverging by speciation events. Ka/Ks values were also calculated among *rhlE1*, *rhlE2*, and *rhlE3* entire nucleotide sequences of 11 different *Burkholderia* species. An average Ka/Ks ratio of ≈1 was obtained, which is usually associated to neutral evolution or to an evolutionary heterogeneity within the gene (Figure 7C). While at the N-terminus these sequences share some conserved regions, including the Q motif or the helicase ATP-binding domain, a high sequence divergence is exhibited among some stretches of the three RhlE-like proteins, namely in the disordered region of the C-terminus. Thus, it seems more plausible that the apparent neutral evolution of the *Burkholderia rhlE*-like genes (Ka/Ks ≅ 1) results from a chimera of positive and purifying selection, rather than from a relaxed selection of these genes. As mentioned before, one of the most obvious differences among these RNA helicases is the sequence and the size of their C-terminal extensions (Figure 7C). These non-conserved extensions are thought to mediate the interactions of DEAD-box proteins with their specific partners or RNA substrates [87].

As already suggested by data in Appendix A, the BCAM0173 helicase was clustered in the same monophyletic group of the *P. aeruginosa* PA0455 and the *E. coli* DbpA (b1343) proteins, and no homologs of the *E. coli* DeaD, RhlB, or SrmB proteins are encoded in the *B. cenocepacia* J2315 genome. However, the BCAM2117 protein, together with other orthologs from species of the β-proteobacteria class were clustered in a monophyletic group relatively close to the cluster that comprises the RhlB and the SrmB proteins (Figure 6). The protein sequence of BCAL2117 only shares an identity of 42.2% and 37.2%, respectively, with the SrmB and the RhlB helicases that are both able to interact with RNase E [79,88].

The degradosomes studied until now comprise at least one DEAD-box RNA helicase [85]. Considering the different pattern of these enzymes in *Burkholderia* species, homologs of components and proteins associated with the RNA degradosome were also searched for within the *B. cenocepacia* J2315 genome. As illustrated in Figure 8, *B. cenocepacia* J2315 possesses two identical RNase E-like proteins, three RhlE-like helicases, and no RhlB homolog was found. Regarding the degradosome associated proteins, in addition to two Hfq-like proteins (mentioned above), three GroEL-like proteins (sharing a protein sequence identity of at least 74%), and two Ppk2 proteins are encoded in the *B. cenocepacia* J2315 genome.

These degradosome associated proteins are encoded as single copies in the *E. coli* genome. The composition of degradosomes varies across bacterial phyla. In *E. coli*, the mentioned helicase RhlB and the RNase E are part of the degradosome, which also includes as major components the exo-ribonuclease PNPase and the metabolic enzyme enolase (Type A, Figure 8). A second type of degradosome was described for *Pseudomonas syringae*, in which RhlE is the major helicase and RNase R is the 3′-5′ exonuclease [89]. To a lesser extent, the SrmB helicase can also be co-purified with RNase E (Type B, Figure 8). A degradosome with a different composition was found in the facultative phototrophic α-proteobacterium *Rhodobacter capsulatus*. The complex comprises RNase E, two ATP-dependent RNA helicases, the transcription termination factor Rho, and a few but not yet identified protein components [90] (Type C, Figure 8). This diversity of mechanisms associated with our data suggest the existence of some peculiarities in *Burkholderia* species regarding global RNA pathways, as the RNA degradation and its regulation.

## 4. Conclusions

RBPs play vital roles in regulating gene expression and cell physiology, although the study of these proteins remains scarce in Bcc bacteria. Considering the limitations of global experimental screens to identify bacterial RBPs, a bioinformatics and comparative genomics strategy was used to identify RBPs encoded within the genome of the opportunistic pathogen *B. cenocepacia* J2315. Many proteins identified in the *E. coli* and *P. aeruginosa* γ-proteobacteria are not present in *Burkholderia* genomes. It is therefore highly likely that unidentified RBPs lacking classic RBDs are encoded in these bacterial genomes. Some evidence on this subject was gathered by running the DeepRBPPred computational model. However, in addition to the majority of genes mentioned in this work, the model also identified several genes with unknown function, as well as genes encoding membrane proteins, leading to no further insights into Bcc RBPs [91]. Nevertheless, in the present study 186 classical RBPs were identified in the *B. cenocepacia* J2315 genome, some of them highlighted by their unusual copy number and/or distinctive percentage of homology. The presence of two distinct Hfq-like proteins, five cold shock-like proteins relatively close to the *E. coli* CspD, three RhlE-like proteins, and multiple copies of several components from the degradosome suggest that in *Burkholderia* species these proteins perform additional functions to those described for the *E. coli* proteins. The results also suggest that the degradosomes of Bcc species have a composition distinct of those from the already described types I, II, and III. In addition to the multiple copies of RhlE-like proteins and degradosome associated proteins that are encoded in Bcc genomes, no RhlB-homolog was found. The information here gathered for the first time on Bcc RBPs, their patterns of expression, and cellular ligands contributes to a better understanding of their contribution to the biology of Bcc bacteria. This information is of critical importance to both understand the strategies used by these opportunistic pathogens when infecting their hosts, as well as to define novel approaches to combat the infections caused by these bacteria. 

## Figures and Tables

**Figure 1 genes-11-00231-f001:**
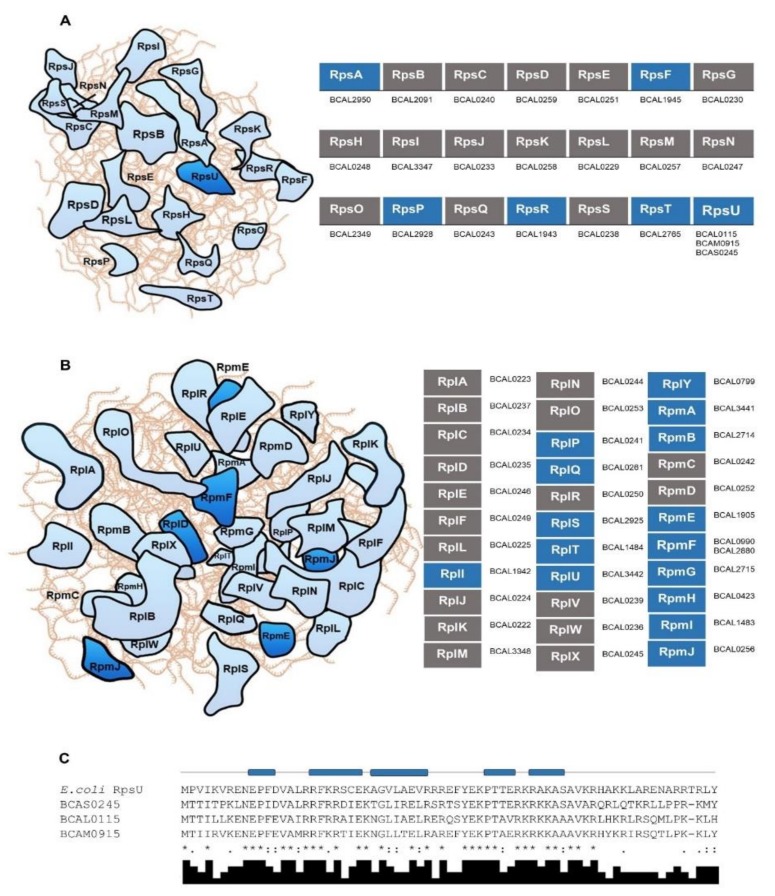
*Burkholderia cenocepacia* J2315 ribosomal proteins that compose the 30S (**A**) and the 50S (**B**) subunits of the bacterial ribosome described for *Escherichia coli*. Panels A and B, left part: schematic representation of the ribosomal RNA (light brown lines) and the r-proteins colored in light blue (one copy gene) or dark blue (at least two paralogs genes in *E. coli* or *B. cenocepacia* J2315). Right part: loci tags of each *B. cenocepacia* J2315 ortholog are mentioned. Rectangles colored in blue represent r-proteins only encoded in bacterial genomes, and grey rectangles represent universal r-proteins that can be found in all domains of life. (**C**) Alignment of the amino acid sequences of the small subunit protein S21 (RpsU)-like proteins from *B. cenocepacia* J2315 (BCAL0115, BCAM0915, and BCAS0245) and *E. coli* (b3065). Asterisks (*) indicate identical amino acid residues, one (.) or two dots (:) indicate semi-conserved or conserved substitutions, respectively. The predicted secondary structure of *E. coli* RpsU protein is shown above the alignment segment, where cylinders represent α-helices. Alignments and secondary structure predictions were performed with MUSCLE [41] and PSIPRED [43] software, respectively.

**Figure 2 genes-11-00231-f002:**
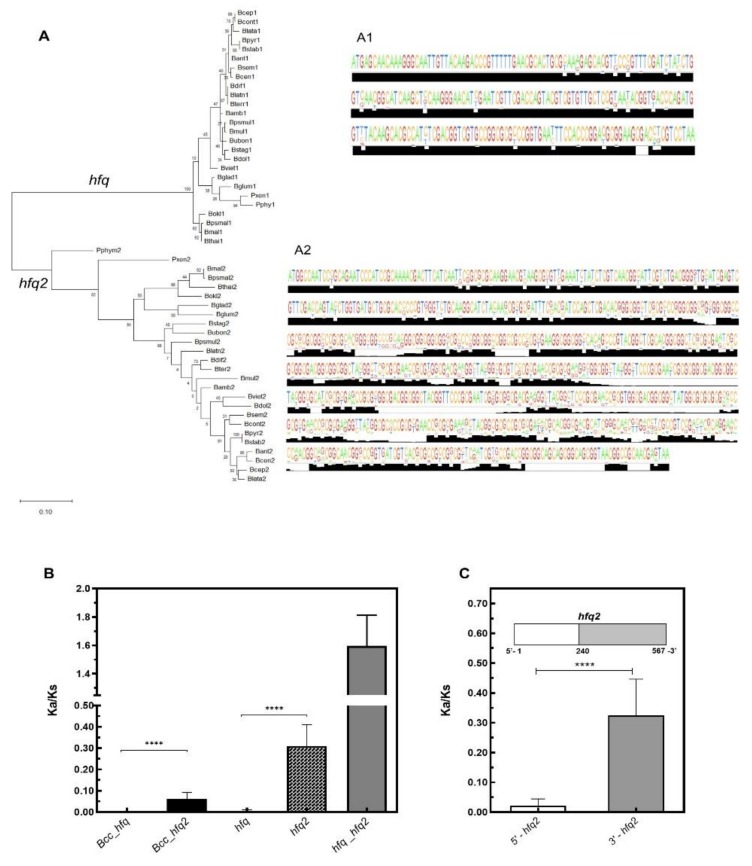
Evolutionary analysis of *hfq* and *hfq2* genes from 24 and 2 species of the genera *Burkholderia* and *Paraburkholderia*, respectively. (**A**) Phylogenetic tree constructed based on the alignment, performed by MUSCLE, of the 52 *hfq* nucleotide sequences listed in Appendix A. The evolutionary relatedness was inferred by using the Maximum Likelihood method and the Tamura–Nei model and conducted in MEGAX software [44]. The tree with the highest log likelihood (−5911.51) is shown. The percentage of trees in which the associated taxa clustered together is shown next to the branches. Initial tree(s) for the heuristic search were obtained by applying the neighbor-joining method to a matrix of pairwise distances estimated using the maximum composite likelihood (MCL) approach. The tree is drawn to scale, with branch lengths measured as the number of substitutions per site. (**A1**) Representation of the consensus sequence, determined by the alignment of *Burkholderia hfq* or (**A2**) *hfq2* genes, as a sequence logo in which the size of each nucleotide and the black histogram corresponds to its degree of conservation. (**B**) Mean Ka/Ks values calculated among the *hfq* and *hfq2* orthologs from the different Bcc or *Burkholderia* bacteria, or among the *hfq* and *hfq2* paralogs from the 26 species selected. (**C**) Mean Ka/Ks values of two distinct regions of the orthologs *hfq2* genes (white for 5′ region and grey for 3′ region). Error bars indicate standard deviation. The *p*-value was determined with the two-tailed Mann–Whitney test and represented by **** when *p* < 0.0001.

**Figure 3 genes-11-00231-f003:**
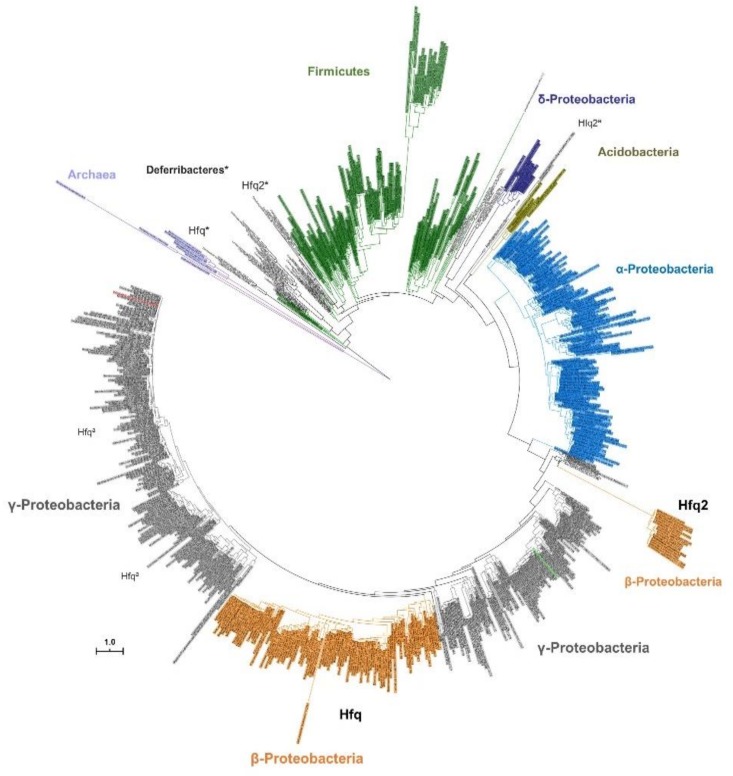
Distribution of Hfq proteins among bacteria and archaea. The phylogenetic tree was inferred by analysis of 798 Hfq protein sequences from 765 distinct species as described in Section 2. The tree is drawn to scale, with branch lengths measured in the number of substitutions per site. Seven distinct groups were formed and highlighted: α-Proteobacteria (blue), β-Proteobacteria (orange), γ-Proteobacteria (grey), Acidobacteria (olive green), δ-Proteobacteria (dark purple), Firmicutes (dark green), and Archaea (light purple). *E. coli* Hfq is colored red and *Pseudomonas aeruginosa* Hfq is green. When present, copies of Hfq-like proteins (Hfq or Hfq2) from γ-Proteobacteria are highlighted with (^a^), while for Deferribacteriaceae these proteins are highlighted with (*).

**Figure 4 genes-11-00231-f004:**
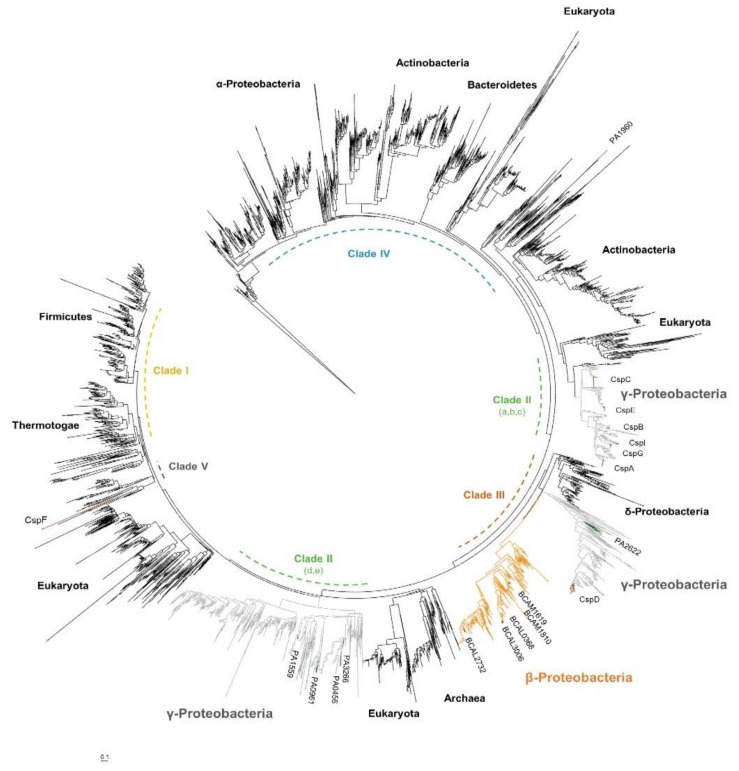
Phylogenetic tree for cold shock proteins of all domains of life. The phylogenetic tree was inferred for the alignment of 4636 non-redundant cold shock protein sequences (annotated with the reference COG1278) from 1436 species. The tree was drawn to scale using the iTol v5 software [45], with branch lengths measured in number of substitutions per site. The groups containing cold shock proteins from γ or β-proteobacteria are colored, respectively, in grey or orange. Cold shock proteins (Csps) from *E. coli* are highlighted in brown and from *P. aeruginosa* in green. The five main clades recently defined by Yu and co-authors are highlighted [64].

**Figure 5 genes-11-00231-f005:**
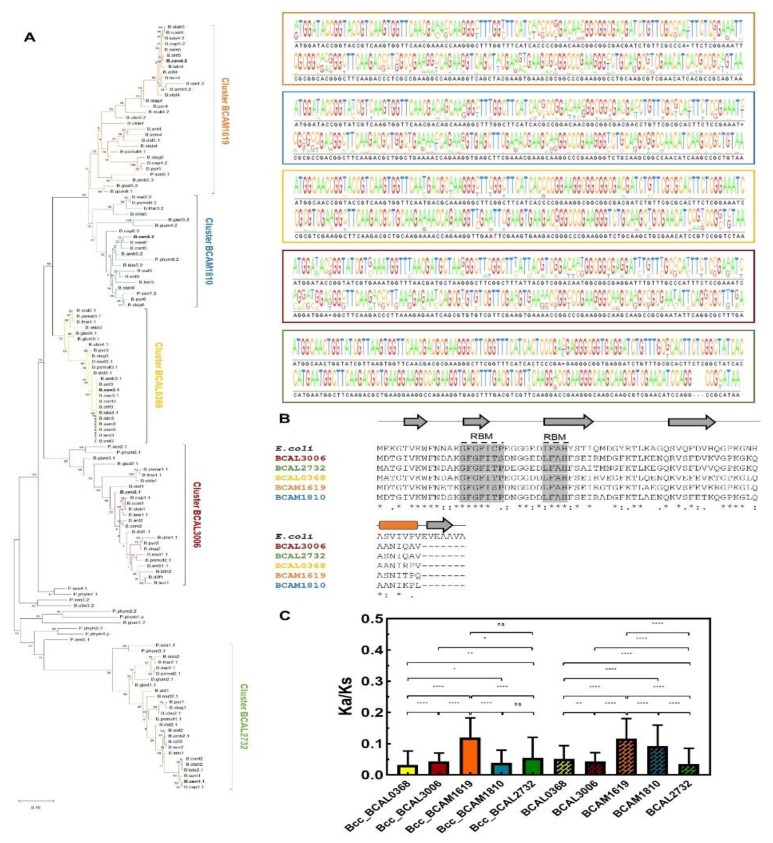
The cold shock-like proteins encoding genes from bacteria of the *Burkholderia* genus. (**A**) Phylogenetic tree constructed based on the analysis of 135 nucleotide sequences listed in Appendix A. The evolutionary relatedness was inferred using the Maximum Likelihood method and Tamura–Nei model and conducted on MEGA X. The tree with the highest log likelihood (−6068.10) is shown. The percentage of trees in which the associated taxa clustered together is shown next to the branches. Initial tree(s) for the heuristic search were obtained by applying the neighbor-joining method to a matrix of pairwise distances estimated using the maximum composite likelihood (MCL) approach. The tree is drawn to scale, with branch length units as the number of base substitutions per site. Upper panels: representation of the consensus sequences, determined by the alignment of the *Burkholderia* Csps genes clustered in Cluster BCAM1619 (orange), BCAM1810 (blue), BCAL0368 (yellow), BCAL3006 (brown), or BCAL2732 (green), as a sequence logo in which the size of each nucleotide with the black background corresponds to its degree of conservation. (**B**) Alignment of the amino acid sequences of the cold shock-like proteins from *B. cenocepacia* J2315 and the *E. coli* CspD (b0880). Asterisks (*) indicate identical amino acid residues, one (.) or two dots (:) indicate semi-conserved or conserved substitutions, respectively. The predicted secondary structure of *E. coli* CspD protein is shown above the alignment segment, where cylinders represent α-helices and arrows represent β-sheets. Alignments and secondary structure predictions were performed with MUSCLE and PSIPRED software, respectively. RBM: RNA binding motif. (**C**) Mean Ka/Ks values of the different Bcc (colored bars without pattern) or *Burkholderia* species (bars with pattern), calculated using orthologs *csps* genes. Error bars indicate the standard deviation. The *p*-values were determined with the two-tailed Mann–Whitney test and represented by * when *p* < 0.05, ** when *p* < 0.01, **** when *p* < 0.0001, or ns—nonsignificant.

**Figure 6 genes-11-00231-f006:**
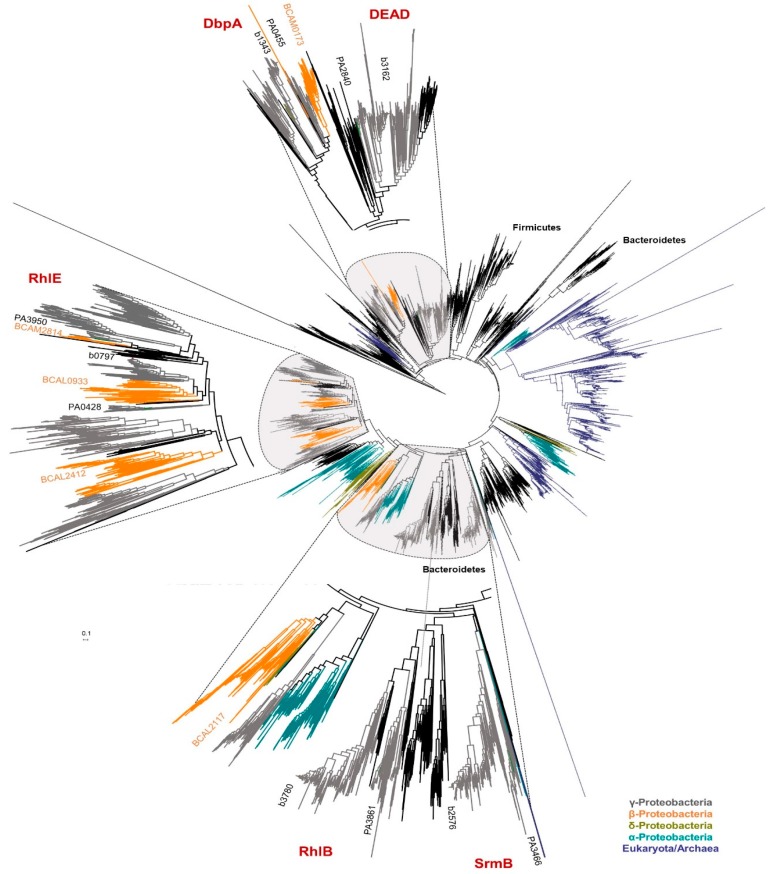
Phylogenetic tree of DEAD-box RNA helicases. The phylogenetic tree was inferred for the analysis of 9254 sequences of proteins annotated with the reference COG0513 from 1815 species. The tree was drawn to scale using the iTol v5 software [45], with branch lengths measured in number of substitutions per site. Clusters containing proteins from bacteria of the proteobacteria class were colored: α-Proteobacteria (blue), β-Proteobacteria (orange), γ-Proteobacteria (grey), δ-Proteobacteria (olive green). Clusters of orthologs genes from DEAD, DbpA, RhlE, RlhB, or SrmB proteins are highlighted. RNA helicases from *E. coli* are colored in brown and from *P. aeruginosa* in green.

**Figure 7 genes-11-00231-f007:**
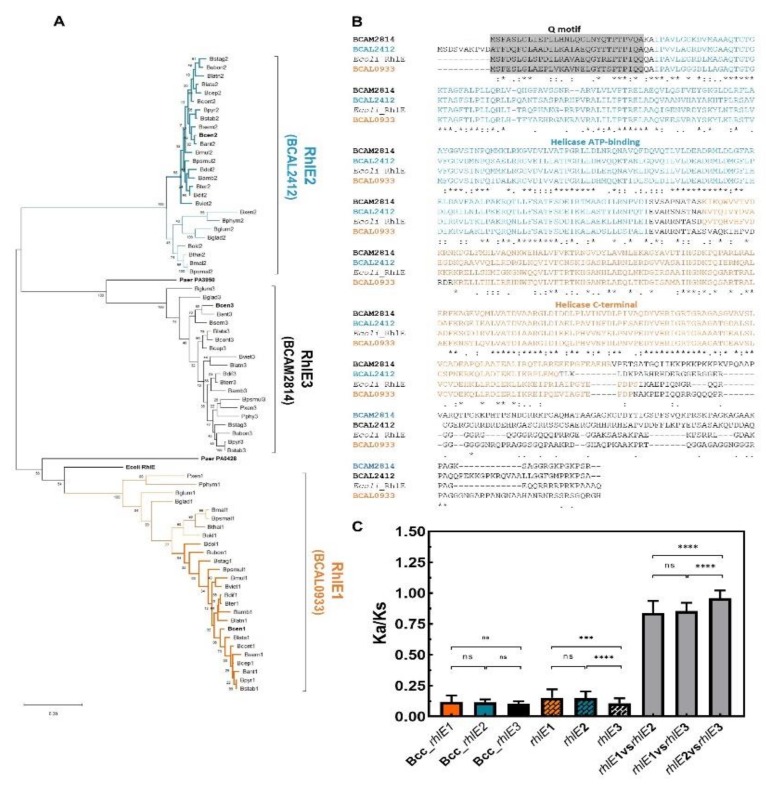
RhlE-like helicases encoded in genomes of *Burkholderia* species. (**A**) Phylogenetic tree constructed based on the alignment of 72 amino acid sequences from 24 species of *Burkholderia* genus and two *Paraburkholderia* species (Appendix A). Sequences of *E. coli* and *P. aeruginosa* RhlE-like proteins were also included. The evolutionary relatedness was inferred using the maximum likelihood method and the Jones-Taylor-Thornton matrix-based model. The tree with the highest log likelihood (−16941.58) is shown. The percentage of trees in which the associated taxa clustered together is shown next to the branches. Initial tree(s) for the heuristic search were obtained by applying the neighbor-joining method to a matrix of pairwise distances estimated using a JTT model. The tree is drawn to scale, with branch length measured as the number of substitutions per site. Evolutionary analyses were conducted in MEGA X. (**B**) Alignment of the amino acid sequences of the RhlE-like helicases from *B. cenocepacia* J2315 and the *E. coli* RhlE. The Q motif is higlighted in grey. Asterisks (*) indicate identical amino acid residues, one (.) or two (:) dots indicate semi-conserved or conserved substitutions, respectively. (**C**) Mean Ka/Ks values of the different Bcc (colored bars) and *Burkholderia* bacteria (colored pattern bars), calculated using orthologs of *rhlE1*, *rhlE2* or *rhlE3* genes and using paralogs *rhlE* genes from 11 *Burkholderia* species (grey bars). Error bars indicate standard deviation. The *p*-value was determined with the two-tailed Mann–Whitney test and represented by *** when *p* < 0.001, **** when *p* < 0.0001 or ns—nonsignificant.

**Figure 8 genes-11-00231-f008:**
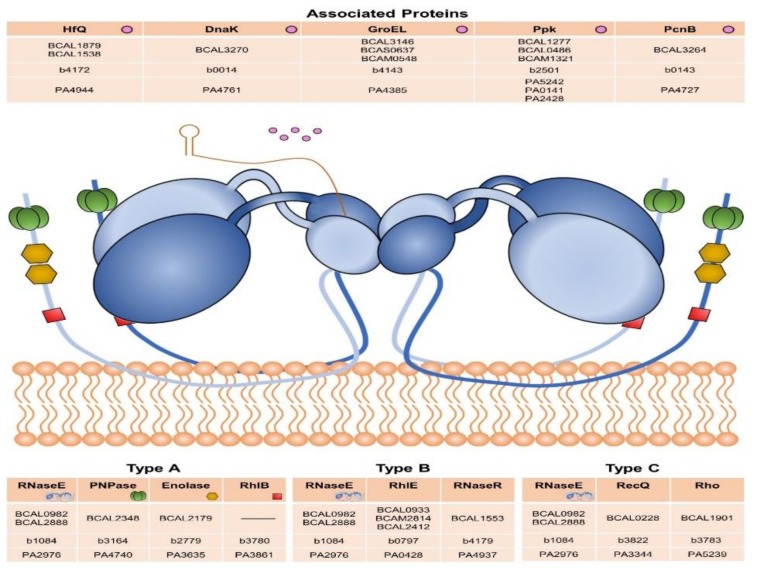
Schematic representation of the membrane-associated RNA degradosome. The main degradosome components were identified and the locus tag of each ortholog gene encoded in *B. cenocepacia* J2315, *E. coli* strain K-12 MG1655, and *P. aeruginosa* PAO1 genomes is mentioned.

**Table 1 genes-11-00231-t001:** Main functions associated with DEAD-box RNA helicases in *E. coli.*

DEAD-Box RNA Helicases	mRNA Processing and Decay	Ribosome Biogenesis	Translation Initiation
DeaD	In vivo formation of a cold-shock degradosome with RNase E [71];In vitro interaction with *E. coli* poly(A) polymerase [72].	Involved in 50S ribosomal subunit assembly, acting after SrmB [73];Putative role in the biogenesis of the 30S ribosomal subunit [73].	Stimulates translation of some mRNAs, probably at the level of initiation [74].
DbpA		Involved in the assembly of the 50S ribosomal subunit [75];RNA-dependent ATPase activity specific for 23S rRNA [76];3′ to 5′ RNA helicase activity that uses the energy of ATP hydrolysis to destabilize and unwind short rRNA duplexes [77].	
RhlB	Component of the RNA degradosome. Interaction with RNase E and co-localization with it at the membrane [78];Facilitates RNase E cleavage of ribosome-free mRNA and highly RNase E-sensitive mRNAs [79];RhlB activity prefers a 5’ single-stranded extension in presence of a fragment of RNase E (aa 628–843) [80]		
RhlE	Interaction with RNase E [79];In vitro interaction with *E. coli* poly(A) polymerase [72];Ability to unwind a short blunt-ended RNA duplex.	May play a role in the interconversion of ribosomal RNA-folding intermediates that are further processed by DeaD or SrmB during ribosome maturation [81].	
SrmB	Stabilization of certain mRNAs [82].Interaction with RNase E [79]In vitro interaction with *E. coli* poly(A) polymerase [72].	Assembly of the 50S ribosomal subunit at low temperature, acting before DeaD [83].

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
