# Peer review of "Comparative Genomics and Evolutionary Analysis of RNA-Binding Proteins of Burkholderia cenocepacia J2315 and Other Members of the B. cepacia Complex"

_genes, 2020, doi:10.3390/genes11020231_

Round 1

Reviewer 1 Report

RNA-binding proteins play critical regulatory roles in bacteria. However, RBPs are poorly defined in non-model organism such as those bacteria from the Bcc group. So the authors of this manuscript mined various databases using the following strategies:

Used keyword search for canonical RNA-binding domains (RBDs) in Pfam/InterPro/Uniprot Identify proteins from E. coli and P. aeruginosa with “RNA binding” related Gene Ontology annotations and search for homologs in B. cenocepacia J2315, confirmed using OrtholugeDB, EggNOG and KEGG

Once the putative RBPs are identified from Burkholderia, the authors performed multiple sequence alignments and phylogenetic analyses, as well as Ka/Ks analysis to look at selective pressure acting on these proteins.

In general, the manuscript is well-written and methodical. I only have minor suggestions and queries:

The methods that the authors chose to utilize are predominately annotation-dependent. We know that even for “model organism” such as E. coli and P. aeruginosa, 1/3 to ½ of the genes have unknown functions/GO/COG annotations. So presumably, what we find in the Bcc would be limited by what we know from E. coli and P. aeruginosa. Given that Bcc typically have very large genome for bacteria and multiple chromosomes, have the authors attempted de novo RBP bioinformatic identification methods such as “RBPPred” (https://academic.oup.com/bioinformatics/article/33/6/854/2557689) or “TriPepSVM” (https://academic.oup.com/nar/article/47/9/4406/5421814)? A quick glance of the supplementary table seems to indicate that almost all of the RBPs (even the various paralogs) are found to be chromosomally-localized. Is this expected?

Author Response

Reviewer #1

COMMENT: RNA-binding proteins play critical regulatory roles in bacteria. However, RBPs are poorly defined in non-model organism such as those bacteria from the Bcc group. So the authors of this manuscript mined various databases using the following strategies:

Used keyword search for canonical RNA-binding domains (RBDs) in Pfam/InterPro/Uniprot Identify proteins from E. coli and P. aeruginosa with “RNA binding” related Gene Ontology annotations and search for homologs in B. cenocepacia J2315, confirmed using OrtholugeDB, EggNOG and KEGG. Once the putative RBPs are identified from Burkholderia, the authors performed multiple sequence alignments and phylogenetic analyses, as well as Ka/Ks analysis to look at selective pressure acting on these proteins.

In general, the manuscript is well-written and methodical.

ANSWER: We appreciate your comments, and evaluation of our manuscript. Thank you.

QUESTION: I only have minor suggestions and queries: The methods that the authors chose to utilize are predominately annotation-dependent. We know that even for “model organism” such as E. coli and P. aeruginosa, 1/3 to ½ of the genes have unknown functions/GO/COG annotations. So presumably, what we find in the Bcc would be limited by what we know from E. coli and P. aeruginosa. Given that Bcc typically have very large genome for bacteria and multiple chromosomes, have the authors attempted de novo RBP bioinformatic identification methods such as “RBPPred” (https://academic.oup.com/bioinformatics/article/33/6/854/2557689) or “TriPepSVM” (https://academic.oup.com/nar/article/47/9/4406/5421814)?

ANSWER: We would like to thank the reviewer for careful reading, and constructive suggestions on the manuscript. Considering the large size of Burkholderia genomes, the comparative analysis performed and their dependence of the existing annotations for the selected model organisms E. coli and P. aeruginosa, we are aware that most RBPs encoded in Burkholderia genomes, and possibly the most interesting, remain unidentified.

Based on this and following your suggestion, the Deep-RBPPred-balance and Deep-RBPPred-imbalance computation models were used to predict and identify new RBPs among the proteome of Burkholderia cenocepacia J2315. Surprisingly, Deep-RBPPred forecasts 2408 (balance model) and 1342 (imbalance model) in a total of 2654 predicted RBPs, when the score cutoff was set to 0.5 as suggested. These values are ridiculously high comparing with the previsions performed for other bacteria (E. coli) and regarding what is currently known about the bacterial RBPs. RBPPred predicts whether a protein binds RNAs integrating the physicochemical properties with the evolutionary information of protein sequences, and the high genome GC content of Burkholderia genomes (>66%) for sure have biased the results and limited the conclusions to be inferred from these results. For these reasons, we decided not to include these results in the manuscript, as the addition of a more than 2000 entry table will add nothing but confusion and dispersion to our work. If required, we can send a file with these results.

Nevertheless, to accommodate this criticism, we have introduced in the “Conclusions” section the following sentences: “Some evidence on this subject was gathered by running the DeepRBPPred computational model. However, in addition to the majority of genes mentioned in this work, the model also identified several genes with unknown function, as well as genes encoding membrane proteins, leading to no further insights into Bcc RBPs.” See new lines 594-597.

QUESTION: A quick glance of the supplementary table seems to indicate that almost all of the RBPs (even the various paralogs) are found to be chromosomally-localized. Is this expected?

ANSWER: Thanks for the observation. In fact, conventional RBPs in Bcc are almost exclusively encoded in chromosomes, i.e, chromosome 1, 2, and 3. There is some discussion concerning if chromosome 3 is indeed a chromosome, and we avoid this controversy. Nevertheless, only 3 RBPs were located in chromosome 3. Burkholderia genomes have several examples of duplicated genes, and we have previously described some of these examples, such as genes encoding acyl carrier proteins (mentioned in the text) and genes encoding bifunctional proteins with phosphomannose isomerase and GDP-D-mannose pyrophosphorylase enzyme activities. We decided not to add any comment to the text concerning the distribution of RBPS between chromosomes and plasmids.

Reviewer 2 Report

The study illustrates the RBPs in Bcc bacteria and its possible mechanisms of RNA regulation and metabolism. Overall the study is straightforward and the experimental design is clear. I only have some minor comments as follows,

The authors should specify why using the model organisms of E. coli and P. aeruginosa as reference organisms. They are both g-proteobacteria, but only P. aeruginosa is known to be prevalent in cystic fibrosis lung disease patients.

L78-79: cystic fibrosis patients – no need to capitalize

L124: no space between key and words

L141: why 50 or 150 bootstrap?

L146: should specify what “ETE” stands for.

Figures 2, 5, and 6 are hard to read. Figures with higher resolution are needed.

Figure 2 The authors specify the strains used for this analysis instead of just saying several strains.

Figure 5 states the bootstrap value of 500 replicates was used. The authors should specify the selected value of bootstrap.

Author Response

REVIEWER #2

The study illustrates the RBPs in Bcc bacteria and its possible mechanisms of RNA regulation and metabolism. Overall the study is straightforward and the experimental design is clear. I only have some minor comments as follows,

QUESTION: The authors should specify why using the model organisms of E. coli and P. aeruginosa as reference organisms. They are both g-proteobacteria, but only P. aeruginosa is known to be first prevalent in cystic fibrosis lung disease patients.

ANSWER: First of all, we would like to thank the reviewer for the thorough revision and criticism of our manuscript and for all comments and suggestions to improve que overall quality of the work. In fact the genomes of two gamma-proteobacteria were used as reference in the present study. P. aeruginosa was used since it is a major cause of lung infections in people with cystic fibrosis and can coexist with Bcc species in the lungs of the patients. E. coli was also included in this analysis since, as for other genes, RBPs are also best studied and characterized in this model organism. Nevertheless, the main reason for using these bacteria as reference is due to the fact that both species possess the best annotated prokaryotic genomes.

A new sentence was introduced in the “Materials and Methods section, which reads as follows: “E. coli and P. aeruginosa were used as reference since both species present the best annotated prokaryotic genomes.” See new lines 101-103.

QUESTION: L78-79: cystic fibrosis patients – no need to capitalize

ANSWER: The proposed changed was performed. Thank you.

QUESTION: L124: no space between key and words

ANSWER: Thank you. The amendment was made.

QUESTION: L141: why 50 or 150 bootstrap?

ANSWER: We apologize for that, there is no fundamental reason why we used 50 and 150 bootstraps. Based on these, new phylogenetic trees (Figure 2, 5 and 7) were constructed using the maximum-likelihood method and 150 bootstraps.

QUESTION: L146: should specify what “ETE” stands for.

ANSWER: ETE is a python toolkit that assists in the automated manipulation, analysis and visualization of any type of hierarchical trees. It provides general methods to handle and visualize tree topologies, as well as specific modules to deal with phylogenetic and clustering trees. To be clearer, we have changed “workflows implemented in the ETE tree building tool” by “workflows implemented in the Python ETE tree building tool”. See new line 147.

QUESTION: Figures 2, 5, and 6 are hard to read. Figures with higher resolution are needed.

ANSWER: We apologize for the quality of the images in this version of the manuscript, but the submitted images had a higher resolution in the document we originally sent. We are uploading a pdf document with the original images.

QUESTION: Figure 2 The authors specify the strains used for this analysis instead of just saying several strains.

ANSWER: Thank you for the suggestion. The proposed changed was performed. See new lines 359-374.

QUESTION: Figure 5 states the bootstrap value of 500 replicates was used. The authors should specify the selected value of bootstrap.

ANSWER: Thanks for the comment. As mentioned above, a new phylogenetic tree was prepared using the maximum-likelihood method and 150 bootstraps. Modifications were carried out in figure legend and in “Materials and methods accordingly. See new lines 142 and 457.

Reviewer 3 Report

Abstract: Title and abstract are very misleading as one needs to read the MS to realize that genome comparison was performed on one Bcc genome and specific genes were selected for other comparisons/analyses.

Introduction: very long and contains arguments that are not touched on in the discussion section. Please shorten.

Methodology:

The methodology section needs more detail and potentially more headlines as it becomes very confusing to understand what was analyzed when reading the result section.

The manuscript has the title ‘comparative genomics of the B. cepacia complex’. Please justify why B. cenocepacia J2315 was selected as a species to investigate as whole genome sequences appear to be available for other Bcc. L126 mentions Bcc species. This is confusing as the headline talks about B. cenocepacia J2315. Is the selection of putative RBPs in other Bcc species based on findings from B. cenocepacia J2315? How many genomes for Bcc are available? A supplemental table might be helpful. ‘Other species of the Burkholderia genus’ how many or are these the same listed in Table S2? L129: Which sequences were downloaded? RBPs? From all Bcc, from how many Burkholderia species, and what is the taxonomic association of sequences from the EggNOG database? Also, the author say ‘OR’; unclear which databases were used for which analysis. L136: Up to 500 sequences is very vague. There are multiple trees in the presented in the manuscript. Be specific which sequences were included for each analysis. L137: ‘default settings’ for both nucleotide and amino acid sequences? L140: Why two different methods to construct trees using different bootstrap values? Only NJ is mentioned in the figure legends.

Discussion:

By choosing a combined results and discussion section, I encourage the authors to start out every section with results. The results set this study apart from any previous study and as such, results should not be buried within the text. In its current state (which could change if more methodological detail is provided), the results/discussion section is very difficult to follow as results presented under specific headlines seem to combine the different analytical strategies chosen by the authors,- single genome comparison and selected genes. Also, throughout the MS it is unclear when the authors refer to Bcc, Burkholderia, or a mixture of both.

L190: first time mentioning of ‘B. pseudomallei …. found in Bcc genomes or in any other strain of the Burkholderia genus’. This analysis is not clear form the methods presented and up to this point only results from B. cenocepacia J2315 were presented/discussed. As a side note, should that be ‘species’ (here and elsewhere)?

L225: the headline is about the Burkholderia genus, but the results are mainly about B. cenocepacia J2315.

L261-277: What is the relevance to the present study?  If gene expression levels at different growth conditions are important to the authors, this section can be condensed to one sentence.

L308: First time mentioning the comparison between hfq2 and hfq, Bcc, and other beta- and gamma proteobacteria. This need to be mentioned in the methodology section. Why including one Paraburkholderia species? L315: 28 Burkholderia and Paraburkholderia species but table S2 shows 23 Burkholderia and 1 Paraburkholderia species? In disagreement with the headline, Figure 3 includes a comparison of more than just beta- and gamma proteobacteria, none of which are mentioned in the methodology.

L341: ‘798 protein sequences’. In the methodology section the authors state ‘up to 500 sequences’.

L352: Part A of the figure does not contribute to the evolutionary analysis particularly when showing an NJ tree. Please remove, which will allow for a larger presentation of panel B/C. Panel B: ‘different Bcc and Burkholderia species’ no mentioning in the methodology section. Panel C: not obvious from Table S2 and why only graphed for hfq2 genes?

L364: Why showing hfg and hfg2 label only for beta proteobacteria? Are these genes contributing to different branches for any of the other taxonomic groups? Even after zooming in 200% E. coli in red and P. aeruginosa in green is not recognizable. Based on the headline it would be better to emphasize on relevant taxonomic representatives.

L384: Parameters selected appear to be random with no justification in the method section.  E.g. L384: 4636 sequences when only max. 500 were mentioned in the method section. L 391: mentioning of clades that are not visible in the graph. L413: Comparison of 24 Burkholderia and 2 Paraburkholderia species when previously selecting less (23 and 1, respectively).

L441: how was evolutionary history inferred from nucleotide sequences using the NJ method? Methodology section describes AA substitutions. Panel C: Based on the main text, I assume 24 Burkholderia species comprising of both Bcc and other Burkholderia species (L414) were investigated to create this graph? Why no separation between Bcc and Burkholderia as shown in figure 2. Why switching to standard error; unclear why the statistical question here would differ from those in Figure 2C and 7C?

L495: From Table S4 it is unclear which 11 species were selected. Also, same comment as above. Why not separated by Bcc and other Burkholderia species? The MS is about the Bcc complex.

L537: degradosomes are a keyword but until the end of the MS there has been no mentioning about them.  

Figure 2A, 5A/B, 7A/B are too small.

Author Response

REVIEWER # 3

QUESTION: Abstract: Title and abstract are very misleading as one needs to read the MS to realize that genome comparison was performed on one Bcc genome and specific genes were selected for other comparisons/analyses.

ANSWER: We would like to thank the reviewer for the thorough revision of the manuscript and all the comments and suggestions. Concerning the work, we stated that we have used the B. cenocepacia J2315 as a reference genome to identify RBPs, and then we extended the analysis to other Bcc and non Bcc members. In order to accommodate the criticisms about the title, we have introduced a change, and it now reads as follows: “Comparative Genomics and Evolutionary Analysis of RNA-binding Proteins of Burkholderia cenocepacia J2315 and other members of the B. cepacia Complex”.We hope that the new title is no longer misleading.

QUESTION: Introduction: very long and contains arguments that are not touched on in the discussion section. Please shorten.

ANSWER: Thanks for the suggestion. Accordingly, we have shortened the introduction by about 15 lines. The introduction now reads as follows, 37-89 :

RNA-binding proteins (RBPs) are found in all domains of life, playing a critical role in the stabilization, protection, processing, and transport of RNA, as well as in the posttranscriptional control of gene expression [1,2]. RBPs are commonly classified based on their specific RNA Binding Domains (RBDs), i.e. structural protein domains that directly bind to specific RNA sequences and/or structured domains in RNA [3]. The classical bacterial RBDs include the S1 domain, the cold-shock domain, the Sm and Sm-like domains, the double-stranded RNA binding domain, the K-homology domain, the DEAD motif, and the ANTAR domain. These domains are widely distributed and/or conserved among different bacterial species (previously reviewed [4,5]). However, proteins that do not harbor any conventional direct RNA-binding site[6], but are able to interact with RNA or proteins in a non-classic way (“unconventional” RBPs) have also been described [7].

Ribosomal proteins (r-proteins) are the most abundant  and best characterized RBPs that have been identified and annotated in bacterial genomes [5,8]. These proteins, together with other RBP major classes such as tRNA synthetases, RNA helicases, and ribonucleases, are critical for many cellular processes. In addition to their involvement in processes associated with RNA metabolism and protein synthesis, the importance of bacterial RBPs in the extensive control of gene expression at the posttranscriptional level has been highlighted over the past two decades. While some RBPs can regulate transcription termination via attenuation (e.g. Rho, NusA and the B. subtilis TRAP and PyrR proteins) or anti-termination mechanisms (e.g. cold shock proteins, HutP, Bgl/Sac), others can repress or activate translation initiation by affecting ribosome biding or by changing RNA stability [9]. The regulation mediated by RBPs is mainly due to their interaction with small non-coding RNAs (sRNAs) [4]. sRNAs are short, non-coding RNA molecules that can act as global regulators of gene expression in prokaryotes [10–13]. In order to perform their regulatory activity, sRNAs often require the aid of global RBPs like RNA chaperones, that facilitate their interaction with cognate mRNA targets, affecting numerous physiological processes [14,15]. Our knowledge of global RBPs remains limited to the Hfq chaperone, the translational repressor CsrA and to the more recently characterized “osmoregulatory” protein ProQ [5,16,17]. This limitation is partially due to experimental difficulties to identify bacterial RBPs, being the advances in understanding these proteins virtually confined to bioinformatics tools to faithfully predict RNA binders in bacteria [18,19]. Moreover, the current knowledge regarding the number, functions and mechanisms of the bacterial RBPs remains also scanty for non-model microorganisms, as is the case of bacteria of the Burkholderia cepacia complex (Bcc). Bcc is a group of at least 24 closely related bacterial species that attracted the attention of various research groups worldwide due to their ability to cause problematic, difficult-to-eradicate, and often fatal infections among cystic fibrosis patients [20–23]. In addition, recent reports also mention an increasing number of infections caused by these bacteria in non-cystic fibrosis patients, including hospitalized patients suffering from other malignancies such as cancer, hemodialysis, and others [24–27]. These bacteria possess large genomes arranged in multiple replicons with high plasticity and complex regulatory mechanisms of gene expression [28]. Our research group has previously reported that 2 distinct Hfq-like RNA chaperones are encoded in the genomes of Bcc bacteria, the 79 amino acid residue Hfq, and the 189 amino acid residue Hfq2 [29].

Besides the Hfq-like proteins, scarce studies are available on RBPs in Bcc bacteria. Therefore, in the present work we report a bioinformatics survey and comparative genomics analyses to identify “conventional” RBPs within the genomes of Bcc. The genome of the highly epidemic strain B. cenocepacia J2315 was used as reference since it is one of the best studied Bcc strains.  The sequences of the identified RBPs, especially those that differed in number of copies per genome, were retrieved from available genome sequences, and their phylogenetic and evolutionary relationships within the Burkholderia genus were analyzed. The predicted functions of these proteins were compared with the E. coli homologs, revealing that in addition to two distinct Hfq-like proteins, five cold shock-like CspD proteins, and three distinct RhlE-like helicases are encoded in B. cenocepacia and in other Bcc genomes. This study contributes to unveil putative protein partners of posttranscriptional regulation in pathogens of the Bcc, suggesting that undisclosed mechanisms should be used by these bacteria to regulate their gene expression.

QUESTION: Methodology: The methodology section needs more detail and potentially more headlines as it becomes very confusing to understand what was analyzed when reading the result section.

ANSWER: Thanks for the criticism. We also agree that this section needs an extra headline. Therefore, we have divided the previous section 2.1 into 2 sections for clarity. These sections now read as follows, in new lines 91-125:

2.1. Database searches for putative RBPs in B. cenocepacia J2315 genome

Putative homologs of RBPs were retrieved from public domain databases using a multipronged search approach. First, the canonic RBDs found widespread among bacterial RBPs were used as keywords in Pfam [30], InterPro [31], and Uniprot databases to search for putative RBPs in B. cenocepacia J2315 genome. The canonical RBDs used were the S1 domain, the cold-shock domain, the Sm and Sm-like domains, the double-stranded RNA binding domain, the K-homology domain, the DEAD motif, the ANTAR domain, the zinc-finger like domain, the PIWI domain, and the PAZ domain.

As a second strategy, proteins from the best known model organism E. coli and the opportunistic pathogen Pseudomonas aeruginosa, classified as “RNA binding” according to Gene Ontology, were used to search for homologs against the proteome of B. cenocepacia J2315. E. coli and P. aeruginosa were used as reference since both species present the best annotated prokaryotic genomes. Briefly, proteins from the taxons E. coli strain K-12 (83333) and Pseudomonas aeruginosa PAO1 (208964) annotated with a RNA binding related GO term were retrieved from the fast web-based browser Quick GO (Gene Ontology Annotation database) [32]. The proteins retrieved were annotated with the GO terms GO:0003723 (RNA binding), GO:0019843 (rRNA binding), GO:0000049 (tRNA binding), GO:1903231 (mRNA binding involved in posttranscriptional gene silencing), GO:0034336 (misfolded RNA binding), GO:0003727 (single-stranded RNA binding), GO:0003729 (mRNA binding), GO:0003725 (double-stranded RNA binding). After obtaining the ortholog genes between E. coli (strain K-12) MG1655 and B. cenocepacia J2315 genomes, or between P. aeruginosa PAO1 and B. cenocepacia J2315 genomes, using the OrtholugeDB predictions (Ortholog Database version 2.1) [33], the protein-coding genes previously selected based on GO terms search were selected and the respective orthologs in B. cenocepacia J2315 genome were retrieved. These data were further confirmed searching for orthologs of each initially selected protein in EggNOG [34] and KEGG [35] databases. The genes sequences retrieved from the above analyses were combined and parsed to remove duplicated genes, to obtain the final list of putative RBPs encoded within the B. cenocepacia J2315 genome.

2.2. Putative RBPs in Bcc and non-Bcc Burkhoderia genomes

Search of orthologs for each selected RBP, whether or not encoded on B. cenocepacia J2315, was performed using KEGG and Burkholderia genomes databases, andthe genomes of Bcc and non Bcc species listed on Table S1.1. [37]. The number of homolog RBPs found among the genomes analyzed, as well as the percentage of Burkholderia species that contain them are shown inTable S1.The phyletic profile of each putative RBP-coding gene listed in Table S1 was obtained based on data provided by OrthoDB [36], the hierarchical catalog of orthologs. Using the RBP names as keywords, the gene orthologs “at Bacterial level” were selected and the evolutionary descriptions were accessed.

The manuscript has the title ‘comparative genomics of the B. cepacia complex’. Please justify why B. cenocepacia J2315 was selected as a species to investigate as whole genome sequences appear to be available for other Bcc.

ANSWER: Thank you for your comment. We changed the manuscript title and we hope that it is no longer misleading. Burkholderia cenocepacia is a threatening nosocomial epidemic pathogen in patients with cystic fibrosis (CF) or with their immune system compromised. Bacteria of this species are highly resistant to available antibiotics, and the strain B. cenocepacia J2315 in particular is the most infectious isolate from CF patients. A new sentence was introduced in the Introduction section to justify centering the analyses on this strain, which now reads as follows: “The genome of the highly epidemic strain B. cenocepacia J2315 was used as reference since it is one of the best studied Bcc strains [23]. See new lines 80-81.

QUESTION: L126 mentions Bcc species. This is confusing as the headline talks about B. cenocepacia J2315. Is the selection of putative RBPs in other Bcc species based on findings from B. cenocepacia J2315? How many genomes for Bcc are available? A supplemental table might be helpful. ‘Other species of the Burkholderia genus’ how many or are these the same listed in Table S2?

ANSWER: Thank you for your comment. As already mentioned above, the point 2.1 was divided into 2 points, 2.1 and 2.2, for the sake of clarity. See new lines 91-125.

The list of RBPs selected from E. coli and P. aeruginosa and listed on Table S1 were used to search for homolog genes in B. cenocepacia J2315 and also in the Bcc and non-Bcc strains listed on Table S1.1, which was added to supplementary data as suggested by the reviewer. See Supplementary Table S1.1.

QUESTION: L129: Which sequences were downloaded? RBPs? From all Bcc, from how many Burkholderia species, and what is the taxonomic association of sequences from the EggNOG database? Also, the author say ‘OR’; unclear which databases were used for which analysis. L136: Up to 500 sequences is very vague. There are multiple trees in the presented in the manuscript. Be specific which sequences were included for each analysis.

ANSWER: Thanks for the comment. We apologize for not explaining more clearly what we have done. Therefore, we have changed the text in order to better explain what was done for each analysis. See new lines 127-149.

QUESTION: L137: ‘default settings’ for both nucleotide and amino acid sequences?

ANSWER: Yes, we have used default settings for both nucleotide and amino acid sequences. This information was added in new line L137.

QUESTION: L140: Why two different methods to construct trees using different bootstrap values? Only NJ is mentioned in the figure legends.

ANSWER: Thanks for the comment. We have new phylogenetic trees (Figures 2, 5 and 7) using the maximum-likelihood method and 150 bootstraps. The legends of each figure was changed accordingly.

QUESTION: Discussion:

QUESTION: By choosing a combined results and discussion section, I encourage the authors to start out every section with results. The results set this study apart from any previous study and as such, results should not be buried within the text. In its current state (which could change if more methodological detail is provided), the results/discussion section is very difficult to follow as results presented under specific headlines seem to combine the different analytical strategies chosen by the authors,- single genome comparison and selected genes. Also, throughout the MS it is unclear when the authors refer to Bcc, Burkholderia, or a mixture of both.

ANSWER: Thanks for the suggestion. The manuscript was revised taking into account the clarity of presentation of results. In some cases it is hard to start a specific topic without mentioning either the methodology or previous results. We cannot understand why it is unclear when we refer to Bcc, Burkholderia or a mix. Bcc means the Burkholderia cepacia complex, a group of species that share in common their ability to cause infections in CF patients. Burkholderia is the genus, and therefore, it includes Bcc and non-Bcc. So whenever we refer to each, the meaning, in our view, is clear.

ANSWER: L190: first time mentioning of ‘B. pseudomallei …. found in Bcc genomes or in any other strain of the Burkholderia genus’. This analysis is not clear form the methods presented and up to this point only results from B. cenocepacia J2315 were presented/discussed. As a side note, should that be ‘species’ (here and elsewhere)?

ANSWER: Burkholderia pseudomallei is a non-Bcc Burkholderia species and is the causative agent of melioidosis. As shown in the new Table S1.1, several strains of this species were included in the analysis performed. We hope the changes made previously can help to clarify this issue. In this particular case, it is appropriate to keep the term strain. Nevertheless, we have checked carefully the use of the terms species or strain throughout the manuscript.

QUESTION: L225: the headline is about the Burkholderia genus, but the results are mainly about B. cenocepacia J2315.

ANSWER: Thanks for the observation. For clarity, the title of 3.2 section was changed: “3.2. R-proteins and other RBPs involved in protein synthesis in the B. cenocepacia J2315 and other bacteria of the Burkholderia genus “

QUESTION: L261-277: What is the relevance to the present study? If gene expression levels at different growth conditions are important to the authors, this section can be condensed to one sentence.

ANSWER: Thanks for the question. We think this information can be relevant, since the expression of these genes under different conditions could be important to the maintenance of three S21 copies in the genomes of these bacteria, as specific genes can be activated or repressed under specific conditions which might differ from each gene encoding essentially for the same function. So, we decided to condense this information in the manuscript.

QUESTION: L308: First time mentioning the comparison between hfq2 and hfq, Bcc, and other beta- and gamma proteobacteria. This need to be mentioned in the methodology section. Why including one Paraburkholderia species? L315: 28 Burkholderia and Paraburkholderia species but table S2 shows 23 Burkholderia and 1 Paraburkholderia species? In disagreement with the headline, Figure 3 includes a comparison of more than just beta- and gamma proteobacteria, none of which are mentioned in the methodology.

ANSWER: Thanks for the observation. Taking into consideration the comment, we have changed the methodology section to clarify the analyses performed. See new lines 127-149.  As mentioned before, the Paraburkholderia genus results from a recent split of the Burkholderia genus in two. Since the “environmental bacteria” were included in the Paraburkholderia genus, we included these bacteria in the analysis performed to detect putative differences that could be related to species virulence.

QUESTION: L341: ‘798 protein sequences’. In the methodology section the authors state ‘up to 500 sequences’.

ANSWER: Thanks for the observation. As suggested, the methodology section was modified to accommodate the criticism. See new lines 142-146.

QUESTION: L352: Part A of the figure does not contribute to the evolutionary analysis particularly when showing an NJ tree. Please remove, which will allow for a larger presentation of panel B/C. Panel B: ‘different Bcc and Burkholderia species’ no mentioning in the methodology section. Panel C: not obvious from Table S2 and why only graphed for hfq2 genes?

ANSWER: We appreciate your comment. A new phylogenetic tree was constructed using the maximum-likelihood method, so we decided to substitute the tree in Figure 2, which helps to visualize the clear separation of the hfq and hfq2-like genes.

The sequences used in panels B and C are listed in supplementary Table S2. For the Bcc Hfq and Bcc Hfq2, only the sequences of Bcc species (highlighted in bold in Table S2) were considered. For the remaining analysis the sequences of all listed species were used. In panel B, the mean Ka/Ks values calculated among hfq and hfq2 paralog genes were also included. Regarding the panel C, only the hfq2 genes were included since the hfq genes are highly conserved over the entire length of the gene, as suggested on panel A and B.

QUESTION:  L364: Why showing hfg and hfg2 label only for beta proteobacteria? Are these genes contributing to different branches for any of the other taxonomic groups? Even after zooming in 200% E. coli in red and P. aeruginosa in green is not recognizable. Based on the headline it would be better to emphasize on relevant taxonomic representatives.

ANSWER: Thanks for the comment. The presence of two distinct hfq-like genes is a characteristic of all Burkholderia and Paraburkholderia genomes sequenced so far and we could not find a similar pattern in the analyses performed. The presence of two copies of this gene among bacterial strains is not common, and excluding Burkholderia species, they were found only in few bacterial species. Among the species analyzed in the phylogenetic tree of Fig 3., four species from the Deferribacteraceae family and 2 gamma-proteobacterial species seems to encode two distinct Hfq-like proteins in their genomes. These distinct copies were highlighted in the phylogenetic tree of Fig.3. We also improved Fig. 3 to make the mentioned details recognizable.

QUESTION: L384: Parameters selected appear to be random with no justification in the method section. E.g. L384: 4636 sequences when only max. 500 were mentioned in the method section. L 391: mentioning of clades that are not visible in the graph. L413: Comparison of 24 Burkholderia and 2 Paraburkholderia species when previously selecting less (23 and 1, respectively).

ANSWER: Thanks for the comment. For phylogenetic trees represented in Figures 3, 4 and 6 we selected all sequences annotated as belonging to the functional categories COG1923 (Conserved Protein Domain family Hfq), COG1278 (Cold Shock Proteins), and COG0513 (Superfamily II DNA and RNA Helicases) in the EggNOG database. This was clarified that in the methodology section. See new lines 142-149.

The Clades mentioned on L391 are now highlighted in new Figure 4.

As mentioned before, the species previously selected for hfq analyses have been replaced by the 24 Burkholderia and 2 Paraburkholderia species used in the further analysis. This information is now more clearly indicated. See new lines 318-320 and 359-360.

QUESTION: L441: how was evolutionary history inferred from nucleotide sequences using the NJ method? Methodology section describes AA substitutions. Panel C: Based on the main text, I assume 24 Burkholderia species comprising of both Bcc and other Burkholderia species (L414) were investigated to create this graph? Why no separation between Bcc and Burkholderia as shown in figure 2. Why switching to standard error; unclear why the statistical question here would differ from those in Figure 2C and 7C?

ANSWER: Thank you for your criticisms, we appreciate and apologize. To respond to the criticism, a new phylogenetic tree was constructed using the maximum-likelihood method.

The 24 species were used to create the graph of panel C. As these cold-shock-like proteins are highly conserved among Burkholderia species, we have only included the analysis comprising all the species listed on Table S4. Following the reviewer suggestion, we performed a new analysis including only the sequences of Bcc species. In the new graphic, the standard deviation was used instead of the standard error.

QUESTION: L495: From Table S4 it is unclear which 11 species were selected. Also, same comment as above. Why not separated by Bcc and other Burkholderia species? The MS is about the Bcc complex.

ANSWER: The 11 species were only used to calculate the Ka/Ks values among the paralog genes to facilitate the process. We have followed your suggestion and a new analysis including only the sequences of Bcc species was performed.

QUESTION: L537: degradosomes are a keyword but until the end of the MS there has been no mentioning about them.

ANSWER: Thank you for your comment. The word “Degradosome” was removed from the Keyword section.

QUESTION: Figure 2A, 5A/B, 7A/B are too small.

ANSWER: We apologize for the quality of the images in this version of the manuscript, but they had a higher resolution in the document we originally sent. During the submission, the images were shortened and lost quality. We are now sending a pdf document with the original images with better quality.

Round 2

Reviewer 3 Report

Several of the figures talk about the evolutionary history inferred from maximum likelihood. Evolutionary history was determined by Ka/Ks ratios. I would recommend to use the term evolutionary relatedness when referring to the trees, as the evolutionary molecular clock was not truly determined by the chosen analysis. Please report highest log likelihood. 

Author Response

Question: Several of the figures talk about the evolutionary history inferred from maximum likelihood. Evolutionary history was determined by Ka/Ks ratios. I would recommend to use the term evolutionary relatedness when referring to the trees, as the evolutionary molecular clock was not truly determined by the chosen analysis. 

Answer: Thanks for the suggestions. The required substitutions were made, in lines 362, 459, and 550, where instead of "evolutionary history" we have substituted by "evolutionary relatedness". 

Question: Please report highest log likelihood.

Answer: In Fig 2, the sentence "The tree with the highest log likelihood (-5911.51) is shown." was added, see new lines 363-364.

In Fig 5, the sentence "The tree with the highest log likelihood (-6068.10) is shown." was added, see new lines 461-462.

In Fig 7, the sentence "The tree with the highest log likelihood (-16941.58) is shown." was added, see new lines 553-554.